# Immune Responses to SARS-CoV-2 Infection and Vaccine in a Big Italian COVID-19 Hospital: An 18-Month Follow-Up

**DOI:** 10.3390/vaccines11010008

**Published:** 2022-12-20

**Authors:** Emanuele Sansone, Carlo Bonfanti, Emma Sala, Stefano Renzetti, Luigina Terlenghi, Alberto Matteelli, Mara Maria Tiraboschi, Tatiana Pedrazzi, Massimo Lombardo, Camillo Rossi, Anna Maria Indelicato, Arnaldo Caruso, Giuseppe De Palma

**Affiliations:** 1Department of Medical and Surgical Specialties, Radiological Sciences and Public Health, Unit of Occupational Health and Industrial Hygiene, University of Brescia, 25123 Brescia, Italy; 2Department of Molecular and Translational Medicine, Institute of Microbiology, University of Brescia—ASST Spedali Civili, 25123 Brescia, Italy; 3Unit of Occupational Health, Hygiene, Toxicology and Prevention, University Hospital ASST Spedali Civili, 25123 Brescia, Italy; 4Department of Infectious and Tropical Diseases, University of Brescia and ASST Spedali Civili, 25123 Brescia, Italy; 5Chief Executive Office, ASST Spedali Civili di Brescia, 25123 Brescia, Italy

**Keywords:** COVID-19, sero-surveillance, vaccination, serological response, healthcare workers

## Abstract

Objectives: This is a longitudinal prospective study which was designed to assess the trend of anti-SARS-CoV-2 antibodies targeting the Spike (anti-S) and Nucleocapside protein (anti-N) viral antigens over a 9-month period after the administration of an anti-SARS-CoV-2 vaccine in a big COVID-19 hospital located in Northern Italy. Participants: 7411 vaccinated workers were included in a linear mixed-effect model analysis performed to model the anti-S decay over the 9 months following the vaccination, during serological screening performed approximately 2, 4, and 9 months following the first jab administration. Serological tests performed in the 9 months preceding vaccine administration were retrospectively analysed to identify the burden of infections occurring before vaccination. Results: The serological assays were used for monitoring the antibody titres during the observational period. Vaccination significantly reduced the rate of infection and elicited a specific humoral response, which lasted during the whole observational period (9 months). A decay was observed in all considered subgroups. At 35 weeks, workers with no history of pre-vaccine infection showed a significantly lower anti-S titre (−2522 U/mL on average (−2589.7 to −2445.7)); younger workers showed significantly higher anti-S titres (140.2 U/mL on average (82.4 to 201.3)). Only seven immunocompromised workers did not show significant levels of anti-S antibodies; three of them, all females, showed a specific T-cell response. Conclusions: Comparing the 9-month periods before and after the first vaccine dose, a significant reduction in infection rate was observed (1708 cases vs. 156). Pre-vaccine infection, especially if contracted during the first pandemic wave, greatly enhanced the response to vaccination, which was significantly affected also by age both in extent and duration (inversely related). A gender effect on the T-cell immune response was observed in a small group of workers who did not produce antibodies after vaccine administration.

## 1. Introduction

Vaccination against SARS-CoV-2 started in Italy, as well as other European countries, on 27 December 2020 (European Vaccine Day), soon after the EMA released a conditional marketing authorization for the BNT162b mRNA vaccine [1]. It had a huge impact on the pandemic course [2], successfully reducing the risk of symptomatic SARS-CoV-2 infections as well as the risk of severe or long COVID-19 [3], hence marking the year 2021 and catching the attention of all scientific community, politicians, and the general population. All health facilities played a pivotal role in tackling the pandemic and their whole workforce (WF, including both healthcare workers (HCW) and non-HCW) were therefore targeted as the first group admitted to vaccination, which was also compulsory for HCW. SARS-CoV-2 sero-surveillance programs were used worldwide to gauge the impact of pandemics on health facility WFs and the general population, to follow the pandemic evolution and estimate the risk of future infections [4,5,6,7].

ASST Spedali Civili is a public, major European COVID-19 centre in Lombardy, Northern Italy, an area strongly hit by the pandemic since its early beginning, which accounted for around 500,000 cases and over 25,000 deaths in 2020 [8]. Before and besides the vaccine, the occupational health surveillance of the hospital WF was based on molecular diagnosis of SARS-CoV-2 infection on rhino-pharyngeal swabs (RPS) performed on suspected cases (symptomatic and/or close contact of confirmed COVID-19 cases) as well as during regular screenings. In 2020, two serological screenings were also performed for epidemiological purposes, the first in spring 2020, as soon as the first serological analysis methods became available, and the second in autumn 2020. Later, the hospital funded the prospective cohort study SeroCoVax-BS, which was mainly aimed at prospectively monitoring the humoral immune response in vaccinated WF. We determined antibodies against the SARS-CoV-2 nucleocapsid (N) protein (anti-N), which are elicited by viral infection, and against the SARS-CoV-2 spike (S) protein (anti-S), which are induced both by infection and vaccine. Such a strategy allowed us, apart from monitoring the serological titre induced by vaccination, to also identify otherwise undetected SARS-CoV-2 infections.

The main objective of the present study was to evaluate the 9-month trend of anti-SARS-CoV-2-S (anti-S) antibody titres in vaccinated workers, also considering age, gender, and pre-vaccine SARS-CoV-2 infections, as well as to evaluate the effectiveness of the vaccination campaign over time. The trend of anti-N antibody titres was also monitored, and anti-N serological conversions in vaccinated workers were used to identify new SARS-CoV-2 infections. Finally, we evaluated the anti-SARS-CoV-2 T-cell response induced by vaccination in a very small group of vaccinated workers who did not develop detectable amounts of anti-S antibodies after two vaccine doses.

## 2. Materials and Methods

### 2.1. Study Design

Figure 1 resumes the main steps and information of the prospective study, also including two previous screenings campaigns that were performed in 2020, the first in spring, evaluating the anti-S IgG response after the first pandemic wave, and then in autumn, evaluating only the anti-N antibody response (Ig G/A/M).

The first SeroCoVax-BS samples for determination of anti-N antibody titres were collected in January–February 2021 (T_0_) and then in March–April 2021 (T_1_), May–June 2021 (T_2_), and August–October 2021 (T_3_) for determination of both anti-S and anti-N antibody titres. All samples collected within 24 h from the vaccine 1st jab were included in the T_0_ group; T_1_, T_2_**,** and T_3_ groups include all samples collected more than 14 days, more than 90 days, and more than 180 days from the vaccine 1st jab, respectively (Appendix A).

A total of 7411 individuals performed at least a test during the observational period. Missing information (i.e., serological test not performed), involved 382 workers at T_1_, 774 at T_2_**,** and 582 at T_3_. Positive T_1_, T_2_, and T_3_ anti-N assays were used to identify new infections that occurred after vaccine administration and therefore assess vaccine effectiveness in previously anti-N negative workers.

### 2.2. Cohort

We planned a prospective longitudinal cohort study involving the WF (mostly HCW) of the ASST Spedali Civili di Brescia, one of the largest tertiary university hospitals in Italy, with over 1500 beds and an estimated workforce of 9436 individuals, including those who are not directly employed but who are continuously working in the hospital. Every worker receiving the 1st dose of the anti-SARS-CoV-2 vaccine (N = 8648 people) had the chance to join the study and undergo a baseline serological screening (T_0_). Further serological blood samples were taken on average after 2 (T_1_), 4 (T_2_), and 8 (T_3_) months. A total of 7411 individuals performed at least a test during the observational period and were enrolled in the study. A full description of the entire cohort, which includes administrative staff, assistant personnel, nurses, physicians, other HCW and “external” workers (employees of external firms), is presented in the Appendix A along with serological test results. Previously infected workers were identified using serological tests performed at T_0_ and in the 9 months before (in spring 2020 and autumn 2020), as they had shown higher sensitivity in detecting previous SARS-CoV-2 infections among our personnel [9] as compared to RPS, that, besides, were not always performed in case of asymptomatic infection. Furthermore, despite the specificity of RPS being particularly high, false positive RPS mainly due to technical errors in sampling or processing can still occur [10]. Since a common characteristic of SARS-CoV-2 infection is the decline in antibody titres [11] and serological reversion of anti-N antibodies has been reported [12], whenever available, the results of serological screenings performed during 2020 were cumulated with the baseline (T_0_) to further minimize the risk of misclassification. New cases of SARS-CoV-2 infection in vaccinated workers were identified by anti-N sero-conversion.

The study protocol and informed consent were approved by the Ethics Committee of Brescia (ID#: NP 4589); written informed consent was obtained from all participants. The study was reported according to the Strengthening the Reporting of Observational Study in Epidemiology (STROBE) Statement.

### 2.3. Patient and Public Involvement

None.

### 2.4. Serological Assays

During spring 2020, serum samples were tested using the chemiluminescent immunoassay Liaison^®^ SARS CoV2 S1/S2 IgG assay (DiaSorin^®^, Saluggia, Italy), whereas, during autumn 2020, electrochemiluminescence immunoassay (ECLIA) Elecsys^®^ Anti-SARS-CoV-2, which detects immunoglobulins (IgG/A/M) anti-N (Roche^®^ Diagnostics International Ltd., Rotkreuz, Switzerland), was used. The response to the vaccine (from T_1_ onwards) was assessed using the ECLIA Elecsys^®^ Anti-SARS-CoV-2 S for anti-S (IgG/A/M) detection (Roche^®^ Diagnostics International Ltd., Rotkreuz, Switzerland).

Liaison^®^ SARS CoV2 S1/S2 IgG is a CLIA assay for the in vitro quantitative detection of IgG anti-S (anti-S1 and anti-S2) in serum and plasma. Recombinant S1 and S2 antigens bound to magnetic beads and the mouse monoclonal antibody anti-human IgG were used to detect and quantitate IgG in human samples. The results are expressed as U/mL, and specimens are considered negative if <12 U/mL, equivocal between 12 and 15 U/mL, and positive if ≥15 U/mL.

Elecsys^®^ Anti-SARS-CoV-2 is an ECLIA immunoassay for the in vitro qualitative detection of antibodies (IgG/A/M) against SARS-CoV-2 in human serum and plasma. The assay uses a recombinant protein representing the nucleocapsid (N) antigen in a double-antigen sandwich assay format. The results are expressed as the cut-off index, the cut-off being 1.

Elecsys^®^ Anti-SARS-CoV-2 is an immunoassay for the in vitro quantitative determination of antibodies (IgG/A/M) to the SARS-CoV-2 Spike (S) protein receptor binding domain (RBD) in human serum and plasma. The assay uses a recombinant protein representing the RBD of the S antigen in a double-antigen sandwich assay format. The results are expressed as U/mL, the cut-off was 0.8 U/mL, and the upper limit of detection was 250 U/mL. Since the antibody titres elicited in immunized individuals were very high, we tested all serum samples at a dilution of 1:20, in accordance with Roche, so the upper limit of detection raised to 5000 U/mL and the dynamic range could be extended.

### 2.5. Anti-SARS-CoV-2 T-Cell Response

The T-cell response against SARS-CoV-2 was evaluated via the QuantiFERON SARS-CoV-2 assay (Qiagen^®^, Venlo, The Netherlands), an interferon-gamma release assay (IGRA) consisting of two antigen tubes that use a combination of proprietary antigen peptides specific to SARS-CoV-2 to stimulate lymphocytes involved in cell-mediated immunity in heparinized whole blood samples. The QuantiFERON SARS CoV-2 Ag1 tube contains peptides targeting CD4+ epitopes from the receptor binding domain (RBD) of the spike protein with overlap to span the entire RBD, whereas the Ag2 tube contains immunodominant CD8+ epitopes selected from the entire spike protein and CD4+ epitopes from the RBD. Plasma from stimulated samples can be used for the detection of IFN-γ using an enzyme-linked immunosorbent assay (ELISA)-based platform. Specimens were processed according to the manufacturer’s guidelines. Following ELISA, quantitative results (IFN-γ concentration in U/mL) were recorded and used for analysis. According to the manufacturer’s instructions, a positive response was defined as a value at least 0.20 U/mL greater than the background U/mL value from the QuantiFERON SARS-CoV-2 Nil tube; the Nil tube value was subtracted to mitigate against background IFN-γ in the sample that was not a result of SARS-CoV-2-specific T-cell stimulation. Median (min–max) Nil subtracted IFN-γ responses were plotted, and the median was chosen to illustrate the central measurement of the dataset in which biological variation could skew results. Minimum and maximum values were provided to inform the range of responses in addition to the central (median) value.

### 2.6. Statistical Analysis

Normality of distributions was assessed using the Kolmogorov–Smirnov test. Categorical variables are presented as frequencies or percentages and compared by the chi-squared test or Fisher’s exact test, as more relevant. Continuous variables were summarised by the means ± standard deviations (SD) when normally distributed or as medians, interquartile range (IQR) when a skewed distribution was observed, and a bootstrapped two-way ANOVA for repeated measures was applied to test differences among groups over time. Multivariate logistic regression was used to identify groups at higher risk of infection. A multivariable linear mixed effect model was performed to estimate the anti-S decay over time adjusted for pre-vaccine SARS-CoV-2 infection, sex, and age to address potential sources of bias and to test group differences through the introduction of interaction terms. Covariates were included based on the hypothesis that they could have an influence on the anti-S trajectories. A restricted cubic spline with 3 knots, in which outer quantiles were set at the 1st and 9th decile, was applied to allow for a non-linear relationship between time and anti-S levels. A total of 500 bootstrap iterations were used to account for the non-normal outcome distribution. The choice of the mixed-effect models also allowed us to deal with missing data at T_1_, T_2_, and T_3_; each subject was considered in the analysis for the time spent in the study contributing to the estimate of the antibodies trend only for the visits the participant attended. The visits where the subjects did not show up were considered as missing at random. All tests were two-sided, and the statistical significance was set at α = 0.05. Analyses were performed through Microsoft-Excel^®^ software, IBM-SPSS^®^ software ver. 26.0.1 (IBM SPSS Inc. Chicago, IL, USA) and R (version 4.1.0).

## 3. Results

Overall, on 6 May 2021, 8648 workers (91.6% of the hospital workforce) were vaccinated against SARS-CoV-2, most of them with the BNT162b2 mRNA vaccine (Comirnaty^®^, Pfizer^®^, New York, NY, USA/BioNTech^®^, Mainz, Germany; >99.1%), and 6862 (79.3%) were included into the T_0_ group of the SeroCoVax-BS Project (Figure 1).

### 3.1. Epidemiologic Evidence of Pre-Vaccine SARS-CoV-2 Infections

Pooling together the serological tests performed in 2020 and those performed at T_0_, the overall number of vaccinated workers with at least one positive serological evidence of a pre-vaccine SARS-CoV-2 infection was 1708 (19.8% of the vaccinated WF), 493 (29%) males and 1215 (71%) females. Table 1 shows the main determinants of pre-vaccine infections: gender, age, and job titles showed some significant effects.

### 3.2. Anti-N Antibody Titres

A significant decrease of anti-N antibody levels over time was observed (median values of 26.5 U/mL at T_0_ vs. 23.1 U/mL at T_1_, 20.1 U/mL at T_2_ and 14.4 U/mL at T_3_) together with a significant increase with age. Higher antibody titres were measured in the older age groups at the different sampling times (*p* = 0.021, Appendix A). We detected 156 anti-N serological conversions from T_1_ onward in not previously infected vaccinated workers, less than half of them (N = 71; 46%) identified by RPS.

During the entire observational period, sero-reversion of anti-N antibody titres was observed in 85 individuals, about 6% of all workers who tested positive for anti-N serological assays. Such event was not affected by gender and was inversely associated with age groups (OR 0.36 (95%CI 0.19–0.69), *p* = 0.002 and OR 0.12 (95%CI 0.02–0.88), *p* = 0.037 for 50–59 years and over 60 years age groups, respectively, compared to the reference age group 20–29).

### 3.3. Anti-S Antibody Titres

At T_1_, anti-S assays demonstrated a 99.9% effectiveness of the vaccine in terms of serological conversion (6 no-responders on 5576 tested, excluding those with pre-vaccine SARS-CoV-2 infection). During the whole observational period, two serological reversions were detected at T_3_ in two over-50 aged males. Table 2 shows that the median anti-S titre in the whole sample was 1458 U/mL (IQR: 774–3063 U/mL). Significantly higher anti-S titres were observed in workers with pre-vaccine SARS-CoV-2 infection (median antibody levels of 5000 U/mL vs. 1157 U/mL, *p* < 0.001). In contrast to anti-N antibodies, an inverse relationship of anti-S antibodies with age groups was observed at T_1_, T_2_, and T_3_ (*p* < 0.001).

The bootstrapped simple linear mixed-effect regression model showed an overall non-linear antibody decay over time that stabilizes at around 1477.4 U/mL (95% CI 1426.7, 1532.1) after 39 weeks since the first jab (Figure 2A). The decreasing trend was significant until around the 30th week after the first dose; in particular, when we fixed the average time of the first interval after the vaccine jab (9 weeks, T_0_–T_1_), we observed an average decrease of −42.3 U/mL per week (95%CI −45.2, −39.5); at the average time of the second interval (18 weeks since the vaccine first jab, T_1_–T_2_), there was a reduction of −37.4 U/mL per week (95%CI −39.5, −35.3) while at the average time of the third interval (35 weeks since the vaccine first jab, T_3_) the decrease per week is almost null (−0.02 U/mL per week, 95%CI −4.1, 4.5).

When introducing an interaction term between time and pre-vaccine infection (yes vs. no; adjusting for age and gender), we observed significant differences at any time between both groups (Figure 3 and Appendix A) as well as different slopes at each time (Appendix A). The decreasing trend was steeper among pre-vaccine infected subjects and the difference in slope increased over time, but after 35 weeks the differences in anti-S antibody titres remained much higher among subjects who were infected before receiving the first vaccine dose (difference at the 35th week: −2522 U/mL; 95% CI −2589.7, −2445.7).

When comparing the trends between genders in the whole sample, we only observed a significant difference during the first weeks (difference at the 9th week: −60.5 U/mL; 95%CI −112.2, −8.5; Figure 4 and Appendix A). No significant difference was observed between the gradient of the two curves, meaning that male and female curves showed similar shapes (Appendix A).

A third comparison was between age groups (above or below 50 years old), always on the sample. In this case, we could observe statistically significant differences at any time, where younger subjects had higher anti-S antibody titres (Figure 5 and Appendix A) as well as differences in slopes with subjects younger than 50 showing a steeper trend until around the 28th week where the two curves showed similar shapes (Appendix A). After 35 weeks, younger subjects showed a higher anti-S antibody titre of 140.2 U/mL on average (95%CI 82.4, 201.3).

When we stratified by gender and pre-vaccine infection groups, we observed significantly higher anti-S antibody titres in females compared to males among subjects that were not infected during the first and last weeks of observation but not in the central part of the follow-up (Figure 6 and Appendix A). Both the curves showed a similar trend (Appendix A). The subjects who contracted a pre-vaccine infection showed a similar response during the first weeks, but the difference in anti-S antibody titres between genders became significant (Figure 6 and Appendix A) because of the males’ steeper decreasing curve (Appendix A).

We then analysed our sample stratifying by age and pre-vaccine infection groups. Among not infected subjects, those below 50 years old showed higher anti-S antibody titres compared to older subjects (Figure 7 and Appendix A). Both curves showed a similar trend (Appendix A). In contrast, the younger subjects who got a pre-vaccine infection showed a similar response to the vaccine compared to older subjects, but then it decreases more rapidly (Appendix A), and we observed a significantly lower anti-S antibody titre (Figure 7 and Appendix A).

We finally compared the subjects with no pre-vaccine infection, those who got an infection during the first wave and those who got a pre-vaccine infection later in time. The last two groups had similar anti-S antibody titres during the first weeks but those who got the infection during the first wave showed higher anti-S antibody titres (Figure 8 and Appendix A) and a less steep slope compared to those with a more recent infection until around week 30 where the difference in trends inverted (Appendix A) but the difference in anti-S antibody titres remained significant also at the end of follow-up (Figure 8).

### 3.4. Anti-SARS-CoV-2 T-Cell Response in Vaccinated Seronegative Subjects

Overall, seven vaccinated workers did not develop a significant amount of anti-S antibodies. All such workers were in treatment with immunosuppressive drugs due to their clinical conditions, which are summarized in Table 3. Six of them were tested with the QuantiFERON SARS-CoV-2 assay approximately 8 months after the vaccination, and three of them (all females) showed T-cell activation but were persistently negative at serological tests (both anti-N and anti-S) as well as at molecular RPS performed every 2 weeks.

## 4. Discussion

To the best of our knowledge, this is the first longitudinal study investigating the trends of anti-N and -S antibody titres in such a large sample for such a long time. Since the beginning of the SARS-CoV-2 pandemic, six in-mass serological screenings were performed on the WF of the ASST Spedali Civili of Brescia Hospital, the last four in the context of the SeroCoVax-BS prospective study. The 2020 serological screenings and those at T_0_ confirmed that our hospital, with a prevalence rate of infected workers of 19.8%, was severely hit by the first (March–April 2020) and second (October–November 2020) pandemic waves. Such a high infection rate can possibly explain the in-mass adhesion (>90%) of workers to the vaccination campaign, well before the enaction of the Italian Law n. 76/2021 on 28 May 2021, which made SARS-CoV-2 vaccination mandatory for HCW. Females showed a significantly lower risk of infection, whereas a higher risk was found both for nurses (OR 1.58; 95% CI 1.29, 1.94, *p* < 0.001) and “other HCW” (OR 1.49; 95% CI 1.20, 1.86, *p* < 0.001), as well as in the youngest age group (age 20–29 years, OR 1.57 95% CI 1.24, 1.98, *p* < 0.001). Vaccination, which mostly occurred with the BNT162b2 vaccine (Pfizer^®^, New York, NY, USA/ BioNTech^®^, Mainz, Germany), succeeded in reducing the rate of infection among our workforce despite the local increase of cases observed during the first trimester of 2021 [13]. A comparison between the first 9 months of the pandemic and those following the vaccination revealed a significant reduction in the infection cases, from 1708 to 156. Vaccination also proved to be very effective in inducing a humoral response: more than 99.9% of tested workers showed positive anti-S antibody titres. A similar rate of effectiveness in eliciting a humoral response was observed in the RENAISSANCE prospective, an observational study on HCW of a large hospital in Milan, Italy, enrolling 2569 workers with no history of previous laboratory-confirmed SARS-CoV-2 infection who completed the BNT162b2 vaccine schedule [14].

Despite the persistence of a positive serological response, our data documented a progressive decay of anti-S antibodies of approximately 46% in 9 months (24% in the T_1_–T_2_ period plus a further 22% during the T_2_–T_3_ interval). In general, the production of anti-SARS-CoV-2 antibodies was strongly influenced by age, showing a complementary pattern: while higher anti-S antibody titres were observed in younger workers, higher anti-N antibody levels were found in the older age groups. The hypothesis that a dysregulated antibody response could be related to the severity of the disease observed in different age groups needs to be further investigated [15]. Nevertheless, the effect of age on humoral response was less evident if compared with those resulting from a pre-vaccine infection. Especially when it occurred during the first pandemic wave, pre-vaccine SARS-CoV-2 infection caused the best vaccine response, with very and persistently high anti-S antibody titres. Serum tests performed on some of these individuals a few days after the first vaccine jab revealed a quick rise in titres up to >5000 U/mL (our unpublished data). This is in accordance with the higher antibody titres observed 14 days after vaccine schedule completion among participants of the RENAISSANCE study with serological evidence of previous SARS-CoV-2 infection [14]. Such findings, also considering the level of in vitro neutralizing activity of anti-S (as low as 15 U/mL) [16], support the hypothesis that these individuals could be particularly protected against COVID-19 after vaccine administration. The most effective response in this subgroup could be related to the vaccine design, which was based on the SARS-CoV-2 Wuhan strain of SARS-CoV-2, or to a more immunogenic response in the case of infection from the latter. The decline in the antibody titres observed over time in vaccinated individuals with a negative history of pre-vaccine SARS-CoV-2 infection is consistent with the waning vaccine effectiveness that has been observed in the general population of Israel [17] and Qatar [18], where both the rates of confirmed SARS-CoV-2 infections and severe COVID-19 cases showed a clear increase as a function of time from vaccination. In comparison, the very low rate of infection observed in our sample in the first 9-months from the first vaccine jab could be explained by a sort of herd immunity in the worker group, possibly resulting from a positive interaction between in-mass vaccination and the complex of protective measures operating in our hospital [19]. In fact, the rate of infections diagnosed via molecular RPS in our sample and that in another hospital in Milan during the 4 months following the vaccination are similar (58/8648 (0.7%) versus 13/2569 (0.5%)) [14].

We also observed a reduction in anti-N antibody titres and 85 cases of sero-reversion in individuals who tested positive at T_0_ or later (T_1_ and T_2_). This phenomenon may lead to misdiagnosis of previous infections. While a positive test for anti-N could be considered a reliable marker of previous SARS-CoV-2 infection, a negative test should not rule it out, especially in younger individuals, who are more prone to lose this type of antibodies over time. This finding again agrees with what was observed in the REINASSANCE study [14]. Only a minority of our workers (7/8648 (<0.1%); the corresponding figure in REINASSANCE being 4/2569 (0.16%), all receiving mycophenolate) did not develop anti-S antibodies after vaccination due to their clinical condition, which required treatment with immunosuppressive drugs. Additional organizing measures were taken to provide them with the highest grade of possible protection; none of such workers became infected during the observational period. They were also offered to test their T-cell response approximately 8 months after the first jab. Three females of the six subjects who accepted to undergo the test showed a positive T-cell response while remaining negative at both serological tests (anti-S and anti-N). Interestingly, those who did not show any T-cell mediated immunity were all males. Our findings corroborate the observation that anti-S antibody levels are significantly higher in females. Males and females are biologically different, and this probably contributes to gender-specific vaccine outcomes. Genetic and epigenetic factors and sex hormones are likely to be involved [20]. Regardless of age, females tend to show greater antibody responses, higher basal antibody levels, and higher B cell numbers than males [20,21]. Furthermore, adult females tend to have higher inflammatory responses and activation and proliferation of T-cells, higher CD4+ T-cells counts, and higher CD4+:CD8+ ratios, whereas males have higher CD8+ T cell frequencies [22,23,24].

Based on the sero-conversion of anti-N antibodies, we could detect 156 new infections during the 9-month follow-up, mostly occurring in the first two months after the first vaccine jab (75 cases, 48%). Such a figure and the constant pauci-symptomatic clinical course of all such cases allowed us to estimate a protection from SARS-CoV-2 infection above 95% and a 100% protection from hospitalization during the first nine months following the first jab administration. A possible limitation of the study is that no anti-S assay was performed at the baseline (T_0_). Misclassification due to possible anti-N sero-reversions was addressed considering the results of serological tests performed during 2020, at the end of the first and second pandemic waves, which were cumulated with the baseline. Preferring serological test results to RPS increased the sensibility and specificity [25] for identifying pre-vaccine infections, hence making the results obtained extremely reliable. The sample size of the cohort, its age heterogeneity, and the duration of follow-up allow to generalize the observed results to similar populations.

## Figures and Tables

**Figure 1 vaccines-11-00008-f001:**
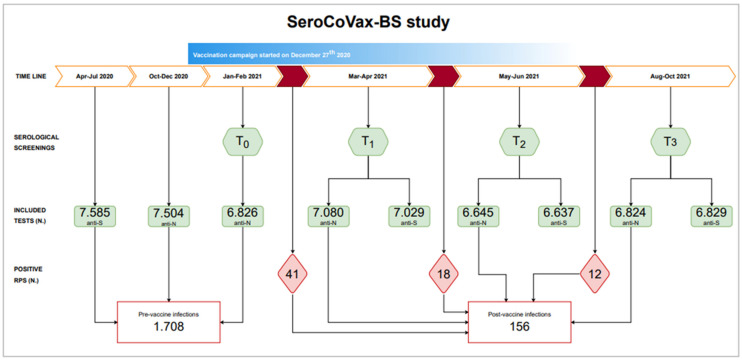
Different phases of the study (green hexagons), number and type of serological assays performed (green rectangles), and number of ongoing infections revealed via rhino-pharyngeal swabs (RPS, red rhombus).

**Figure 2 vaccines-11-00008-f002:**
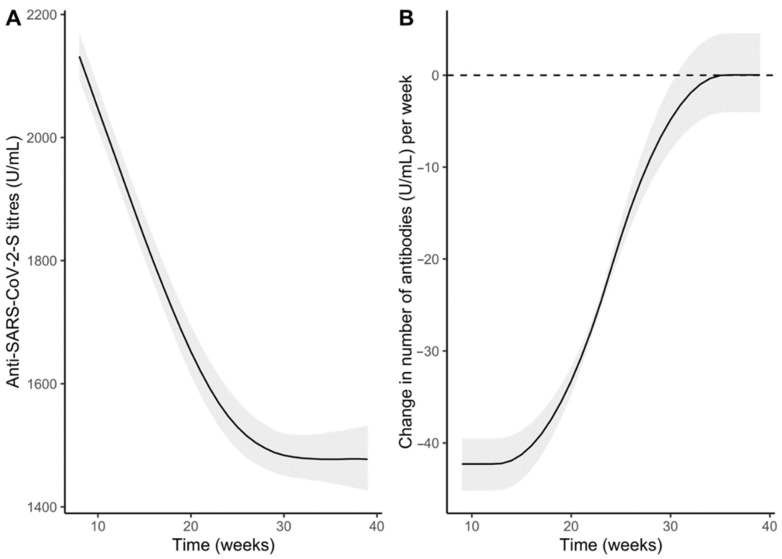
Overall trend of the anti-S antibody titres (U/mL) over time (**A**) and its gradient (**B**). Curves were obtained from the predictions of the bootstrapped simple linear mixed model.

**Figure 3 vaccines-11-00008-f003:**
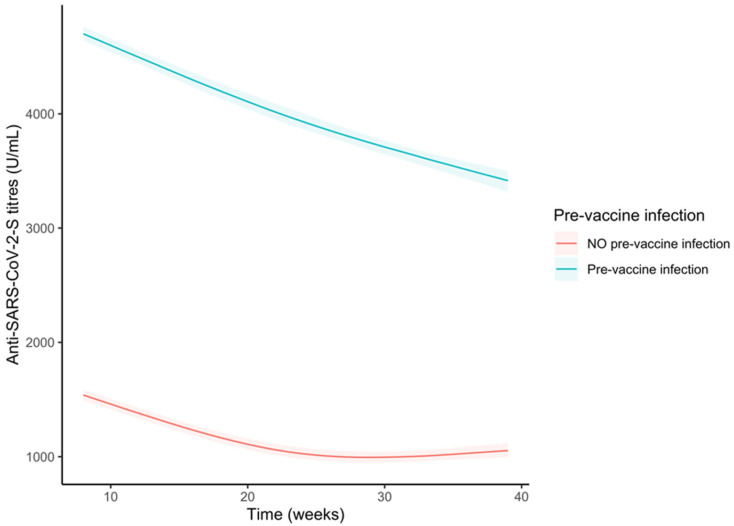
Trends of the anti-S antibody titres (U/mL) over time in the whole sample stratified by pre-vaccine SARS-CoV-2 infection. Curves were obtained from the predictions of the bootstrapped linear mixed model adjusted by age and sex.

**Figure 4 vaccines-11-00008-f004:**
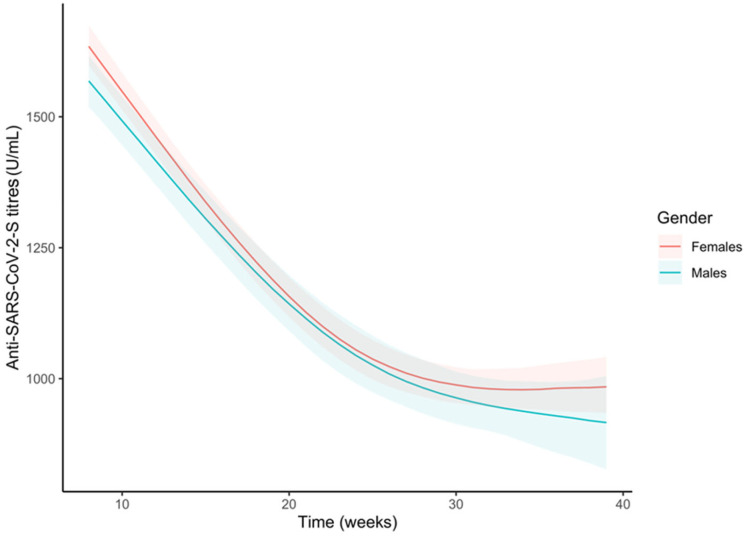
Trends of the anti-S antibody titres (U/mL) over time by gender in the whole sample. Curves were obtained from the predictions of the bootstrapped linear mixed model adjusted by age and pre-vaccine SARS-CoV-2 infection.

**Figure 5 vaccines-11-00008-f005:**
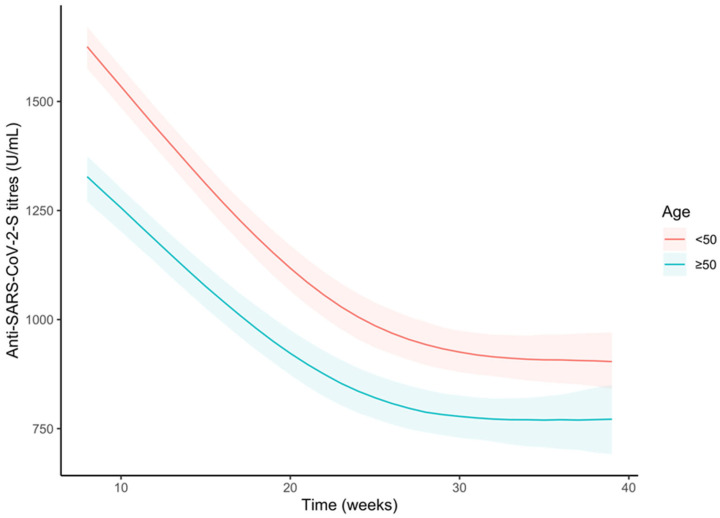
Trends of the anti-S antibody titres (U/mL) over time by age group (<50 years old or ≥50 years old), in the whole sample. Curves were obtained from the predictions of the bootstrapped linear mixed model adjusted by sex and pre-vaccine SARS-CoV-2 infection.

**Figure 6 vaccines-11-00008-f006:**
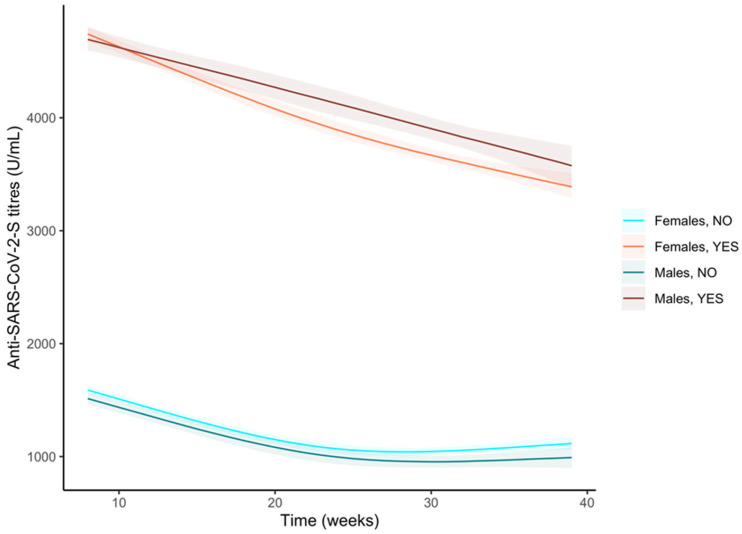
Trends of the anti-S antibody titres (U/mL) over time in the sample stratified by pre-vaccine SARS-CoV-2 infection and gender. Curves were obtained from the predictions of the bootstrapped linear mixed model adjusted by age.

**Figure 7 vaccines-11-00008-f007:**
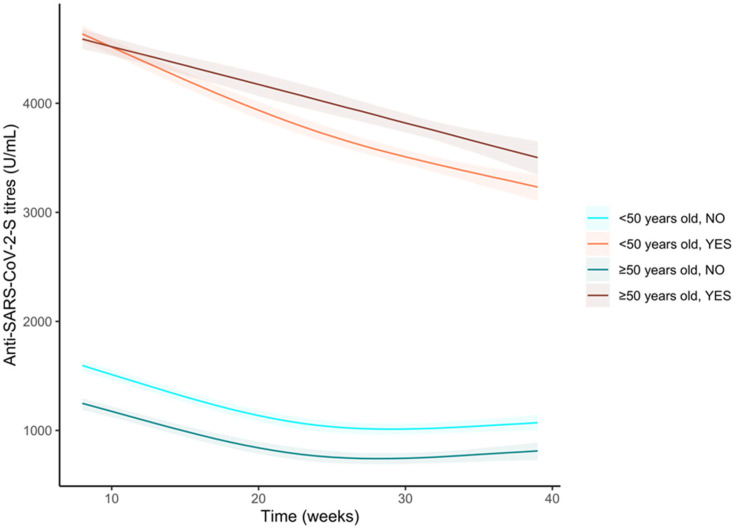
Trends of the anti-S antibody titres (U/mL) over time in the sample stratified by pre-vaccine SARS-CoV-2 infection and age groups (<50 years old or ≥50 years old). Curves were obtained from the predictions of the bootstrapped linear mixed model adjusted by sex.

**Figure 8 vaccines-11-00008-f008:**
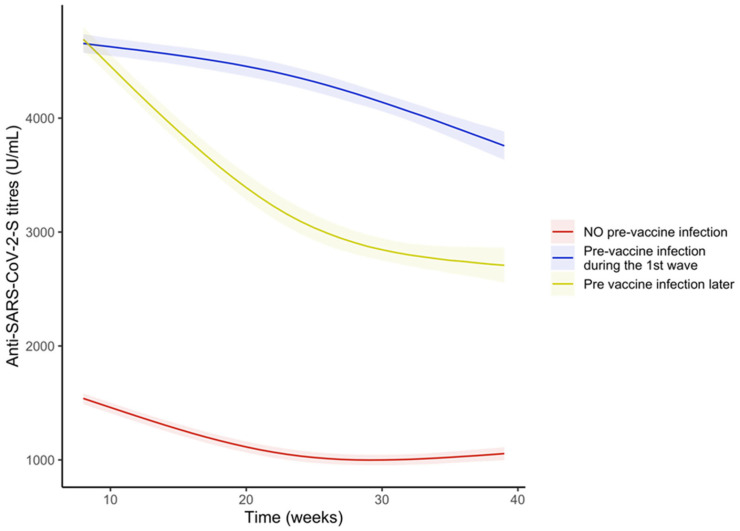
Trends of the anti-S antibody titres (U/mL) over time in the sample stratified by pre-vaccine SARS-CoV-2 infection, categorized in three groups: no pre-vaccine SARS-CoV-2 infection, pre-vaccine SARS-CoV-2 infection occurred during first wave and pre-vaccine SARS-CoV-2 infection occurred after first wave. Curves were obtained from the predictions of the bootstrapped linear mixed model adjusted by age and sex.

**Table 1 vaccines-11-00008-t001:** Distribution of pre-vaccine SARS-CoV-2 infections by gender, age group, and job title. The results of the Fisher or chi-squared tests and the multivariable logistic regression (odds ratios (OR) and 95% confidence intervals (95% CI)) are shown. HCW: healthcare workers.

Variables	N.	Pre-Vaccine Infection Before T_0_ (N = 1708)	*p* Value *	OR (95% CI) **	*p* Value **
Gender					
Male	2404	493 (20.5%)	0.278	1 (reference)	0.002
Female	6244	1215 (19.5%)		0.83 (0.73–0.93)	
Age					
over 60 years	730	122 (16.7%)	<0.0001	1 (reference)	
50–59 years	2659	520 (19.6%)		1.13 (0.91–1.41)	0.267
40–49 years	1992	397 (19.9%)		1.14 (0.91–1.43)	0.267
30–39 years	1845	328 (17.8%)		1.07 (0.85–1.35)	0.554
20–29 years	1422	341 (24.0%)		1.57(1.24–1.98)	<0.001
Job title					
Administrative	885	140 (15.8%)	<0.0001	1 (reference)	
Technician	656	128 (19.5%)		1.28 (0.98–1.67)	0.073
Other HCW	1500	323 (21.5%)		1.49 (1.20 1.86)	<0.001
Nurse	2648	615 (23.2%)		1.58 (1.29 1.94)	<0.001
Physician	2428	393 (16.2%)		0.95 (0.76–1.18)	0.613
External workers	531	109 (20.5%)		1.30 (0.98 1.72)	0.067

* Fisher’s exact test or Pearson’s chi-squared test. ** Logistic regression model, including gender, age group, and job title as covariates.

**Table 2 vaccines-11-00008-t002:** Distributions of anti-S antibody titres (U/mL; medians and first and third quartiles) in workers stratified by age groups and pre-vaccine SARS-CoV-2 infection across time (at T_1_, T_2_, and T_3_). *p*-values are estimated through the bootstrapped ANOVA for repeated measure.

Variables	T_1_ (N = 7029)	T_2_ (N = 6637)	T_3_ (N = 6829)	*p*
Overall	1458 (774–3063)	1103 (609–2171)	792 (428–1664)	<0.001
Age groups				<0.001
20–29 years	2300.5 (1355–5000)	1770 (1121–3582)	1244 (779–2482)	
30–39 years	1660 (959–3165)	1216 (733–2249)	858 (492–1709)	
40–49 years	1298.5 (702–2818)	980 (563–1964)	713 (398–1543)	
50–59 years	1238 (665–2566)	950 (513–1862)	669 (356–1427)	
over 60 years	1034 (561–2173)	763.1 (433–1535)	551 (305–1108)	
Pre-vaccine SARS-CoV-2 infection				<0.001
Yes	5000 (5000–5000)	5000 (3781–5000)	4059 (2016–5000)	
No	1157 (671–1900)	896 (529–1435)	631 (369–1078)	

**Table 3 vaccines-11-00008-t003:** Main characteristics of workers not showing any humoral response (anti-S antibodies) to the vaccine.

Gender	Age	Disease	Immunosuppressive Drugs	T Cell Assay
Male	45	Transplant recipient	Unspecified	Negative
Male	55	Acute lymphoblastic leukemia; Transplant recipient	Unspecified	Negative
Male	57	Transplant recipient	Unspecified	Negative
Female	31	Wegener granulomatosis	Rituximab, prednisone	Positive
Female	47	Multiple Sclerosis	Ocrelizumab	Positive
Female	52	Neuromyelitis optica	Rituximab	Positive
Female	47	Transplant recipient	Unspecified	Not performed

## Data Availability

Data subject to third-party restrictions: Data are available from the authors, with the permission of the tertiary hospital ASST Spedali Civili di Brescia.

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
