# Peer review of "Immune Responses to SARS-CoV-2 Infection and Vaccine in a Big Italian COVID-19 Hospital: An 18-Month Follow-Up"

_vaccines, 2022, doi:10.3390/vaccines11010008_

Round 1

Reviewer 1 Report

the authors present an interesting prospective longitudinal study on the response to COVID 19 vaccination.

There are various grammatical errors.

It would be interested if the authors include in their study the percentage of patients who had side effects to the administration of the vaccine.

should cite these manuscripts:

DOI: 10.3390/vaccines10020308

DOI: 10.1002/rmv.2318

    DOI: 10.1001/jamaneurol.2021.2004

    A second study could also be performed in children.

    Author Response

    Reviewer's comment 1#: There are various grammatical errors.

    Author's response : We are somewhat surprised about such criticism, because revisor 2 on the contrary claims: " It is a pleasure to read. Impeccable English."  Anyway the paper has undergone a careful english language editing. 

    Reviewer's comment 2#: It would be interesting if the authors included in their study the percentage of patients who had side effects following the administration of the vaccine.

    Author's response: We thank the reviewer fo the comment, but such aspect was outside our main objectives. Possibly we will include this aspect in a further study on vaccine's side effects.

    Reviewer's comment 3#: should cite these manuscripts:

    DOI: 10.3390/vaccines10020308

    DOI: 10.1002/rmv.2318

      DOI: 10.1001/jamaneurol.2021.2004

      Author's response: We thank the reviewer fo the comment but the citations are outside the main objective of the study (see the above answer). 

      Reviewer's comment 4#: A second study could also be performed in children.

      Author's response: We thank the reviewer for the comment, that however is beyond our actual possibilities.

      Reviewer 2 Report

      This is an outstanding research project of great value, well planned, executed, analysed and presented. It is a pleasure to read. Impeccable English.

      My comments are minor and for clarity, and to aid future researchers.

      It would be great if you could capture your key conclusions in a box

      You wrote: “Missing information (i.e. serological test not performed), involving 382 workers at T1, 774 at T2 and 582 at T3 was addressed with statistical analysis”

      [can you please state in detail your method and how your procedure affects the results]

      “Every worker receiving the 1st dose of the anti-SARS-CoV-2 vaccine (N= 8.648 people) had the chance to join the study and undergo a baseline serological screening (T0).“

      [workforce 9436, participated 8648, Table 2 T1 7024, T2 6829. How do these numbers affect your conclusions? Agree, it is difficult to get participation from thousands of workers on different shifts, sick leave and holidays]

      “Furthermore, despite the specificity of RPS being particularly high, false positive RPS mainly due to technical errors in sampling or processing can still occur”

      [please comment on false positive rate for future researchers]

      “At T-2, serum samples were tested using the chemiluminescent immunoassay Liaison SARS CoV2 S1/S2 IgG assay (DiaSorin, Saluggia, Italy), whereas, after July 2020 (T-1), electrochemiluminescence immunoassay (ECLIA) Elecsys® Anti-SARS-CoV-2, which detects immunoglobulins (IgG/A/M) antiN (Roche Diagnostics International Ltd, Rotkreuz, Switzerland) was used. The response to the vaccine (from T1 onwards) was assessed using the ECLIA Elecsys® Anti-SARS-CoV-2 S for anti-S (IgG/A/M) detection (Roche Diagnostics International Ltd, Rotkreuz, Switzerland).”

      [Please comment how changes in methods could affect results, for future researchers]

      “Mixed effect models also allowed to deal with missing at-random outcome data”

      [please comment on methods and outcomes]

      ----

      Typo in Table 3: Wegener not Wegner

      Author Response

      Author’s responses to reviewer’s 2# comments

      Reviewer's comment 1#: It would be great if you could capture your key conclusions in a box

      Author's response: A summary box will be possibly included in the updated draft.

      SUMMARY BOX proposal for the editor
      Section 1: What this study adds.
      Pre-vaccine infection greatly enhances the response to vaccination, especially if contracted during the first pandemic wave (SARS-CoV-2 Wuhan strain).
      Antibody titres induced by the vaccine are significantly affected by age (inversely related) in extent and duration.
      In a small group of people not producing antibodies after vaccine administration, a T-cell immune response is observed only among females.
      Section 2: Strengths and limitations of the study.
      This is the first longitudinal study investigating the trends of anti-N and -S antibody titres in a so large sample for a so long time.
      The sample size of the cohort, its age heterogeneity, and the duration of follow-up allow to generalize the observed results to similar populations.

      Reviewer's comment 2#:

      You wrote: “Missing information (i.e. serological test not performed), involving 382 workers at T1, 774 at T2 and 582 at T3 was addressed with statistical analysis”

      [can you please state in detail your method and how your procedure affects the results]

      Author's response: Thanks for highlighting this. Since this is not the statistical analysis section we rephrased the sentence as followMissing information (i.e. serological test not performed), involved 382 workers at T1, 774 at T2 and 582 at T3”. We added more details in the statistical analysis section: “The choice of the mixed effect models also allowed us to deal with missing data at T1, T2 and T3: each subject was considered in the analysis for the time spent in the study contributing to the estimate of the antibodies trend only for the visits the participant attended. The visits where the subjects did not show up were considered as missing at random.

       Reviewer's comment 3#:

      “Every worker receiving the 1st dose of the anti-SARS-CoV-2 vaccine (N= 8.648 people) had the chance to join the study and undergo a baseline serological screening (T0).”

      [workforce 9436, participated 8648, Table 2 T1 7024, T2 6829. How do these numbers affect your conclusions? Agree, it is difficult to get participation from thousands of workers on different shifts, sick leave and holidays]

      Author's response: We agree that missing values are always an issue in longitudinal studies. However, among the 7.411 enrolled workers we did not observe a high dropout rate (5.2% at T1, 10.4% at T2 and 7.9% at T3) and we can consider the reason for dropout as at random since they are not related to the outcome. This should prevent us from getting biased estimates.

      Reviewer's comment 4#:

      “Furthermore, despite the specificity of RPS being particularly high, false positive RPS mainly due to technical errors in sampling or processing can still occur”

      [please comment on false positive rate for future researchers]

      Author's response: Thank you for highlighting the error. The missing reference will be included in the updated draft. (Healy, B, Khan, A, Metezai, H, Blyth, I, Asad, H. The impact of false-positive COVID-19 results in an area of low prevalence. Clin Med (Lond) 2021;21(1):e54–e56.CrossRefGoogle Scholar)

      Reviewer's comment 5#:

      “At T-2, serum samples were tested using the chemiluminescent immunoassay Liaison® SARS CoV2 S1/S2 IgG assay (DiaSorin, Saluggia, Italy), whereas, after July 2020 (T-1), electrochemiluminescence immunoassay (ECLIA) Elecsys® Anti-SARS-CoV-2, which detects immunoglobulins (IgG/A/M) antiN (Roche Diagnostics International Ltd, Rotkreuz, Switzerland) was used. The response to the vaccine (from T1 onwards) was assessed using the ECLIA Elecsys® Anti-SARS-CoV-2 S for anti-S (IgG/A/M) detection (Roche Diagnostics International Ltd, Rotkreuz, Switzerland).”

      [Please comment how changes in methods could affect results, for future researchers]

      Author's response:

      Thank you for highlighting this point. Before vaccination serological screening were meant to identify all natural infections occurred and gauge the impact of the pandemic on our workforce. Changes in method did not affect the classification of pre-vaccine infection because all the tests performed before vaccination were considered as binary variables (positive or negative) and their results were cumulated. Indeed, a direct comparison between the anti-S antibody levels observed at the end of the 1st pandemic wave (induced by natural infection, Liaison®) and those following the vaccination (Elecsys®) was not performed because of such changes. In multicentric studies involving centers using different titer scales, comparison has been made using normalized log-transformed antibody level and SD as scale unit (for instance: Front. Immunol., 29 September 2022; Sec. Vaccines and Molecular Therapeutics). Finally, all the results collected after vaccination were obtained using the same serological test (Elecsys®).

       Reviewer's comment 6#:

      “Mixed effect models also allowed to deal with missing at-random outcome data”

      [please comment on methods and outcomes]

      Author's response: Please see the answer to comment 2. We replaced the mentioned sentence with the one in the answer to comment 2.

      Reviewer's comment 7#: Typo in Table 3: Wegener not Wegner

      Author's response: Thank you for highlighting the error. It has been fixed.